# Cyclosporiasis—Updates on Clinical Presentation, Pathology, Clinical Diagnosis, and Treatment

**DOI:** 10.3390/microorganisms9091863

**Published:** 2021-09-02

**Authors:** Blaine A. Mathison, Bobbi S. Pritt

**Affiliations:** 1ARUP Laboratories, Institute for Clinical and Experimental Pathology, Salt Lake City, UT 84108, USA; blaine.mathison@aruplab.com; 2Division of Clinical Microbiology, Mayo Clinic, Rochester, MN 55905, USA

**Keywords:** *Cyclospora*, protozoan, diarrhea, diagnostics, pathology, parasite, parasitic

## Abstract

*Cyclospora cayetanensis* is an intestinal coccidian parasite transmitted to humans through the consumption of oocysts in fecally contaminated food and water. Infection is found worldwide and is highly endemic in tropical and subtropical regions with poor sanitation. Disease in developed countries is usually observed in travelers and in seasonal outbreaks associated with imported produce from endemic areas. Recently, summertime outbreaks in the United States have also been linked to locally grown produce. Cyclosporiasis causes a diarrheal illness which may be severe in infants, the elderly, and immunocompromised individuals. The increased adoption of highly sensitive molecular diagnostic tests, including commercially available multiplex panels for gastrointestinal pathogens, has facilitated the detection of infection and likely contributed to the increased reports of cases in developed countries. This manuscript reviews important aspects of the biology, epidemiology, and clinical manifestations of *C. cayetanensis* and provides an in-depth discussion of current laboratory diagnostic methods.

## 1. Introduction

Cyclosporiasis is a foodborne and waterborne intestinal parasitic disease caused by the coccidian parasite *Cyclospora cayetanensis*. While other *Cyclospora* species have been described from non-human primates [1], *C. cayetanensis* is the only species known to infect humans and to date has only been isolated from humans, although isolates from captive chimpanzees and macaques in Europe have been found to have isolates genetically similar to *C. cayetanensis* [2].

*Cyclospora cayetanensis* occurs worldwide, with hot spots of endemicity including tropical and subtropical regions of Latin America (including the Caribbean), Central and Southeast Asia, the Middle East, and North Africa [3,4]. This parasite is particularly prevalent in settings with poor sanitation where the environment is contaminated with human feces from infected individuals. Cyclosporiasis exhibits varied seasonality worldwide, which may be affected by precipitation, temperature, and humidity [3].

Isolated cases in developed countries are usually from individuals returning from endemic areas [4]. As surveillance for cyclosporiasis has improved in recent years, seasonal outbreaks are becoming commonplace in many parts of the world. In the last 10 years, summertime outbreaks have been documented in Canada [5,6,7,8], Poland [9], the United Kingdom [7], and the USA [10,11,12,13,14,15,16,17]. The sources of outbreaks are usually fresh produce that is typically eaten raw, especially plants that grow low to the ground and are prone to being exposed to fecally contaminated water and soil. Common vehicles implicated in outbreaks include raspberries, blackberries, strawberries, blueberries, basil, cilantro, snow peas, snap peas, and various lettuces [3,4]. Often, the source of the outbreak is not known, because fresh produce has a short shelf life and the products are usually consumed or expired and discarded before an outbreak is realized [3].

According to the U.S. Centers for Disease Control and Prevention (CDC), there have been approximately 6000 domestically acquired cases of cyclosporiasis in the past three years [18]. Outbreaks in the US have historically been associated with produce imported from Latin America. However, the US Food and Drug Administration (FDA) detected the first evidence of *C. cayetanensis* in locally grown produce (cilantro) in 2018 [3]. Since then, *C. cayetanensis* has been increasingly detected in domestic food and surface water [18], likely due in part to improved surveillance tools. Given the ongoing annual outbreaks of cyclosporiasis in the United States, the FDA formed the *Cyclospora* Task Force in 2019, and this group produced the “*Cyclospora* Prevention, Response and Research Action Plan” to combat foodborne illness associated with imported and domestically grown produce [19].

## 2. Biology and Life Cycle

*Cyclospora cayetanensis* has a complex life cycle involving both sexual and asexual development within a single host. Infection is initiated by the ingestion of fully sporulated oocysts in fecally contaminated food or water. The oocysts excyst in the lumen of the small intestine, and sporozoites invade the epithelial cells lining the duodenum and jejunum. The sporozoites become trophozoites, which in turn become either Type I meronts (schizonts) or Type II meronts. Type I meronts contain 8–12 merozoites and perpetuate autoinfection in the host. Type II meronts each contain four merozoites, which go on to form microgametocytes (microgamonts) and macrogametocytes (macrogamonts) to initiate the sexual cycle. A microgametocyte fertilizes a macrogametocyte, resulting in the formation of a zygote. Zygotes become oocysts in the enterocytes and are shed in an unsporulated state in the feces [3,4,20]. Oocysts sporulate in the environment, at which time they become infective to other people. Factors affecting sporulation in nature are still unresolved, but the process may be influenced by humidity, soil chemistry, and exposure to ultraviolet light. Under laboratory conditions, sporulation takes approximately 7–14 days at 22 °C and 30 °C [21] (Figure 1).

*Cyclospora cayetanensis* is usually confined to the upper small intestine in immunocompetent hosts, but it can cause ectopic infection of the biliary tree and gall bladder in patients with HIV infection and AIDS [22,23,24].

## 3. Pathogenesis

Parasite invasion and replication within enterocytes damages the small intestinal epithelium, leading to the disruption of the brush border, loss of membrane bound digestive enzymes, and intestinal villous blunting and atrophy [25,26]. An influx of lymphocytes, plasma cells, and occasionally eosinophils occurs in the lamina propria. These changes have the overall effect of decreasing the small intestinal absorptive capacity, leading to decreased uptake of water, nutrients, and electrolytes [25].

## 4. Clinical Presentation

The presentation of infection varies with the age and immune status of the host, as well as the local endemicity of infection [25]. Infection is often mild or asymptomatic [4], particularly in residents of highly endemic countries. When present, symptoms include profuse watery diarrhea, abdominal cramping, nausea, fatigue, low-grade fever, anorexia and weight loss. Less commonly, mucus or blood may be found in the stool. More severe disease occurs most commonly in infants, the elderly, and profoundly immunocompromised patients such as those with HIV/AIDS [25]. Travelers from non-endemic countries are also likely to experience severe infection. Prolonged diarrhea can result in dehydration and malnutrition and may rarely result in death, particularly in infants and individuals with other infections or morbidities.

Symptom onset usually occurs after a median incubation period of seven days following the ingestion of infectious oocysts (with a range from 2 to ≥2 weeks) and may last for weeks to months without treatment [27,28]. Some patients experience a single self-limited episode, whereas others have waxing and waning symptoms [29,30].

Rarely, *C. cayetanensis* may infect the biliary tract and cause acalculous cholecystitis [23,24], particularly in immunocompromised hosts. Guillain-Barré syndrome, ocular inflammation, reactive arthritis, and sterile urethritis have also been reported [25].

## 5. Treatment

Trimethoprim/sulfamethoxazole (TMP/SMX; trade names Bactrim, Cotrim, Septra) is the treatment of choice for cyclosporiasis [31,32]. It is administered at a dose of one double-strength (DS) 160 mg/800 mg tablet given orally twice per day for 7–10 days and has been shown to provide >90% cure rates in immunocompetent patients. The efficacy of TMP/SMX for treating cyclosporiasis was first demonstrated in a placebo-controlled trial of 40 adult expatriates and tourists in Nepal [29]. The authors found that only 1 of 16 patients (6.3%) had detectable oocysts in stool after seven days of treatment with TMP/SMX, compared with 15 of 17 patients (88.2%) who received a placebo. Importantly, an improvement in symptomatology was correlated with the eradication of oocysts. Nitazoxanide or ciprofloxacin are recommended for patients that are unable to take TMP/SMX due to sulfa allergy, although treatment failure may occur [32,33].

Profoundly immunocompromised patients such as those with AIDS and transplant recipients may require a longer course of treatment and/or a higher dose of TMP/SMX. Ongoing prophylaxis is also recommended to prevent relapse [34]. A 1994 study of HIV-positive adults in Haiti found that symptomatic infection recurred in 12 of 28 patients (43%) who were monitored for more than one month after a 10 day course of TMP/SMX given orally four times per day [34]. All responded promptly to repeat treatment and were subsequently given TMP/SMX three times a week for secondary prophylaxis; of these, only one patient recurred after seven months. These authors published a follow up study in 2000 showing that patients with HIV were successfully treated with seven days of TMP/SMX and DS tablets given orally twice per day, followed by prophylaxis for 10 weeks (DS tablet given orally, three times per week). Regardless of the initial dose, these studies clearly show the importance of prophylaxis for preventing relapse. Based on these data, the 2019 guidelines from the American Society of Transplantation recommend a 10 day course of TMP/SMX (one DS tablet given orally four times per day) for solid organ transplant recipients, followed by secondary prophylaxis with TMP/SMX (one DS tablet given orally three time per week) [35]. The reduction of immunosuppression is also indicated if possible.

There is currently no vaccine for cyclosporiasis. Instead, preventative measures focus on improving sanitation (e.g., measures to prevent human feces from entering the environment and contaminating the food and water supply) and treating food to inactivate contaminating oocysts. The oocysts are highly resistant to commonly used disinfectants but are inactivated by cooking. Travelers to highly endemic areas are advised to avoid eating uncooked raw vegetables and unpeeled fruits and preferentially to choose foods that are fully cooked and served hot [35]. Similarly, patients with HIV and solid organ transplant recipients should avoid consuming untreated well and surface water to avoid gastroenteric infections [36,37].

## 6. Diagnosis

The diagnosis of intestinal parasites is laborious, time-consuming, and often requires specialized expertise [38]. Still, ova-and-parasite (O&P) exams and other forms of stool microscopy are routinely ordered for patients presenting with diarrhea and other intestinal manifestations, even when other diagnostic methods may be more appropriate. In developed counties, if a parasitic disease is suspected in an immunocompetent patient with diarrhea and no travel history to endemic areas for parasitic diseases, parasites such as *Giardia duodenalis* and *Cryptosporidium* spp. should be considered before ordering O&P exams [38]. In the United States and Canada, where cyclosporiasis has become a seasonal illness in the summer, *C. cayetanensis* should also be considered as a primary differential in any patient presenting with compatible symptoms and illness onset during the cyclosporiasis peak period (i.e., May through August). A history of consuming fresh leafy greens, berries, basil, and cilantro within 2 weeks prior to the onset of illness may raise the clinical suspicion for cyclosporiasis, although this type of nutritional history is not commonly obtained [12]. Cyclosporiasis should also be considered in patients returning or emigrating from areas endemic for the disease, in which case specialized assays such as modified acid-fast (MAF) stain, safranin stain, and UV autofluorescence (see below) should be ordered to compliment the routine O&P exams [38]. Except during seasonal outbreaks, cyclosporiasis is rarely considered by a health care provider, and *C. cayetanensis* may be overlooked when only routine O&P examinations are ordered. Importantly, it may be necessary to examine multiple stool specimens for *C. cayetanensis* to make a diagnosis of cyclosporiasis, as the number of oocysts shed in stool may be relatively few.

Unfortunately, while there are numerous rapid antigen-detection assays for *G. duodenalis* and *Cryptosporidium* spp., the diagnosis of cyclosporiasis still relies heavily on stool microscopy (Table 1). Only recently have nucleic acid amplification tests (NAATs) started to become available, and even then, options are limited and tests may be cost prohibitive. To date, there are no antibody or antigen detection assays for the routine clinical diagnosis of cyclosporiasis.

## 7. Stool Microscopy

Like other coccidians and *Cryptosporidium*, *C. cayetanensis* can present a challenge for clinical microbiologists and parasitologists. It is not always easily detected in the standard O&P examination used for most intestinal parasites, in part because the unsporulated oocysts have few diagnostic morphologic features that set them apart from clinically insignificant objects in stool, and they do not take up the routinely used iron hematoxylin and trichrome stains [20]. For microscopic analyses, specialty stains, such as modified acid-fast and safranin, as well as ultraviolet autofluorescence, are used to enhance detection [20].

## 8. Wet Mounts

Wet mounts can be performed on concentrated or unconcentrated stool specimens preserved in an appropriate medium for parasite diagnostics. The traditional O&P examination employs a two-vial system containing polyvinyl alcohol (for trichrome and iron hematoxylin staining) and 10% buffered formalin (for wet mount preparation); however, there are numerous single-vial systems available that are suitable for both preparations, and many use more environmentally safe fixatives [38].

In concentrated wet mounts of stool, oocysts of *C. cayetanensis* are 8–10 µm in diameter. They possess a thick oocyst wall and central morula that contains 4–6 refractile globular internal structures; because the oocysts are shed unsporulated, sporozoites are not present in freshly passed stool (Figure 2A) [39]. Diagnosis by wet mount microscopy can be enhanced by the use of differential interference contrast (DIC) (Figure 2C) or ultraviolet (UV) autofluorescence (Figure 2B) [4]. DIC has the advantage of enhancing the oocyst wall and internal structures but is not readily available in most clinical laboratories.

Autofluorescence is a natural biological phenomenon in which light of a longer wavelength is emitted when an object is illuminated with light of a shorter wavelength [40], and the method can greatly improve the sensitivity of the detection of *C. cayetanensis* oocysts in unstained wet mounts. When viewed under UV light using an excitation filter of 330–365 nm, the oocyst wall of *C. cayetanensis* will fluoresce blue (Figure 2B). A less intense green fluorescence can be observed using a blue excitation of 450–490 nm [4,20,40]. Oocysts observed with a 510–530 nanometer barrier filter, as commonly used for the detection of pathogenic fungi with calcofluor white, will appear green [40]. It is important not to add iodine to the wet mount (which is commonplace for the O&P examination) as it will interfere with the fluorescence. The oocysts and sporocysts of other coccidians, such as *Cystoisospora belli* and *Sarcocystis* spp., will also autofluoresce [20,41], as will the eggs of several helminths, including *Enterobius vermicularis*, *Trichuris trichiura*, *Ascaris lumbricoides*, hookworm, and *Hymenolepis nana* (Mathison, unpublished data). Given its increased sensitivity over permanent smears, such as MAF or safranin, labs should consider adding a UV screen whenever *C. cayetanensis* is suspected.

Lacto-phenol cotton blue (LPCB) has also been evaluated for the detection of coccidians and *Cryptosporidium* in concentrated wet mounts of stool. While results were not optimal for the detection of *Cryptosporidium*, they were promising for the detection of *C. cayetanensis* and *C. belli*, suggesting that LPCB may be a good screening tool for coccidians in resource-poor areas where acid-fast staining is not available [42].

## 9. Permanent-Stained Smears

Oocysts of *C. cayetanensis* do not stain with the commonly used stains employed for permanent smears (i.e., iron hematoxylin and trichrome) and appear as colorless refractile spheres that may be wrinkled or collapsed and have a “ground glass” appearance when viewed over multiple focal planes (Figure 2D).

The detection of *C. cayetanensis* in permanent-stained smears can be enhanced by the use of modified Zieh–Neelsen (ZN), cold Kinyoun’s MAF, or modified safranin stain. When the oocysts take up the stain with ZN and MAF, they appear pink to red with a wrinkled appearance (Figure 2E). Unfortunately, a large percentage of oocysts might not successfully take up the stain, resulting in “ghost forms” that appear similar to those stained with trichrome (Figure 2E) [4,20,38,39,41]. Staining with the hot safranin method results in more uniform staining of oocysts (Figure 2F), but it is often not preferred as it requires the heating of the stain [20,38,40]. A microwave can be used to facilitate the heating step [43]. Oocysts of *C. cayetanensis* stained with safranin appear pink to red or orange and have a wrinkled appearance, similar to what is seen with the MAF and ZN methods [20,38].

Auramine O (auramine-phenol) has also been used for the detection of *C. cayetanensis* and related organisms, although there is disagreement on the efficacy of this method [38,40]. In two studies in Egypt, the authors concluded that Auramine O was superior to ZN and MAF because of the more consistent staining of the oocysts [44,45]. However other studies, including those in Haiti [34] and the United Kingdom [46], suggest that the sensitivity is too low to be reliable for diagnosing *C. cayetanensis*. In addition, the auramine–phenol method requires the use of a fluorescent microscope, while the ZN, MAF, and safranin methods do not, which could add extra expense to the testing algorithm and may be more challenging in resource-poor regions [38].

## 10. Histopathology

*Cyclospora cayetanensis* can be identified in biopsy specimens of the small intestine; however, it should be noted that endoscopy and biopsy are not routinely ordered for the diagnosis of cyclosporiasis, as several more sensitive and less invasive diagnostic alternatives exist. Most commonly, *C. cayetanensis* is found in histologic sections of intestinal biopsies in situations in which the cause of symptoms is unknown and non-infectious etiologies are included in the differential diagnosis, thus necessitating a broad diagnostic work-up.

In histologic sections, *C. cayetanensis* parasites are seen within cytoplasmic parasitophorous vacuoles in the apical aspect of the intestinal epithelial cells, above the nucleus, and are usually most numerous at the tips of the villi. There is usually associated villous flattening and increased numbers of chronic inflammatory cells in the lamina propria. Diagnosis can usually be made using routine hematoxylin and eosin staining (Figure 3). Acid-fast stains provide little additional benefit for histologic diagnosis; pre-oocyst stages will not usually be highlighted with ZN and Fite acid-fast stains, since the organism is only partially acid-acid fast and then only in the oocyst wall. In biopsy specimens, *C. cayetanensis* may be difficult to separate from *C. belli*; however, the parasitic forms of the latter are larger and occur in the basal aspect of the epithelial cells, below the nucleus [47,48].

## 11. Molecular Diagnosis

Compared to other groups of clinically relevant microorganisms, the molecular diagnosis of parasitic diseases is less commonly employed but is quickly gaining traction (Table 2). The molecular diagnosis of *C. cayetanensis* in stool specimens is primarily done through the use of multiplex assays that contain multiple bacterial, viral, and parasitic targets, although large reference labs may have their own laboratory-developed tests (LTDs).

The BioFire^®^ FilmArray^®^ Gastrointestinal (GI) Panel (Biomérieux, Lyon, France) has both FDA clearance for in vitro diagnostic use in the United States and is Conformitè Europëenne (CE) marked. At the time writing, it is the only FDA-approved multiplex NAAT that can detect *C. cayetanensis*. The FilmArray GI Panel has 22 targets, including three additional parasites: *Giardia duodenalis*, *Entamoeba histolytica*, and *Cryptosporidium* spp. A study during a large outbreak in the Midwestern United States in 2018 demonstrated the efficacy of the FilmArray GI Panel in the successful diagnosis of cyclosporiasis [49]. The authors also noticed an increase in the number of *C. cayetanensis* cases reported nationally in the U.S. between 2011 and 2018 and proposed that the implementation of the FilmArray GI Panel may be responsible, at least in part, for the increased detection of this organism [49]. A similar observation was made during an outbreak in Iowa and Nebraska in 2013 [50].

Among the multiplex NAATs that are CE marked but not FDA cleared are the Allplex™ Gastrointestinal Panel (Seegene, Seoul, South Korea), a multiplex assay with 23 targets including six parasites (*C. cayetanensis*, *Blastocystis* sp., *Cryptosporidium* spp., *Dientamoeba fragilis*, *E. histolytica*, and *G. duodenalis*) [51] and the QIAstat-Dx^®^ Gastrointestinal Panel (Qiagen, Hilden, Germany), which has 21 targets including four parasites (*C. cayetanensis*, *Cryptosporidium* spp., *E. histolytica*, and *G. duodenalis*) [51]. Similarly, the EasyScreen™ Enteric Protozoan Extended Detection Kit (Genetic Signatures, Newtown, Australia) has six protozoan and microsporidial targets, including *C. cayetanensis*. While there do not seem to be any publications yet evaluating the efficacy of the extended version of this assay, a smaller version of this assay (EasyScreen™ Enteric Protozoan Detection Kit) showed 92–100% sensitivity and 100% specificity for the parasite targets it includes (five targets, does not include *C. cayetanensis*) [52]. Finally, the Novodiag^®^ Stool Parasites test (Mobidiag, Espoo, Finland) is a relatively new CE-marked test that combines real-time PCR and microarray assays for the rapid and fully automated diagnosis of stool pathogens. There does not appear to be any literature available yet evaluating this technology, but according to their website, the Novodiag^®^ Stool Parasites assay has 26 parasite targets, including *C. cayetanensis* [53].

The use of molecular tools could speed up the recognition of an outbreak. Because outbreaks are usually associated with ephemeral food sources (raw, fresh fruits and vegetables), by the time an outbreak is realized, the source may have been consumed or discarded, making trace-back investigations difficult [50]. NAATs also do not require parasitology expertise, and some are available in an easy-to-use, cartridge-based format that provide a result in as little as one hour. However, despite the increased sensitivity and specificity afforded by NAATs, there are some disadvantages associated with molecular diagnosis. In general, NAATs are still the most expensive option for routine clinical diagnosis and require sophisticated instruments and proprietary reagents. Furthermore, some of the commercially available tests, as well as most LDTs, for *C. cayetanensis* must be performed in a high complexity laboratory with strict containment controls and unidirectional workflows. Finally, it is important to note that NAATs can detect DNA from both viable and dead organisms and thus should not be used as a test of a cure, as DNA may continue to be shed in the stool for some time after successful treatment [51]. Similarly, the detection of *C. cayetanensis* DNA does not necessarily indicate that the organism is the cause of the patient’s illness, as infection with this parasite may be asymptomatic and there may be another responsible pathogen. Finally, specimens submitted for NAAT may not be compatible with additional tests such as microscopy and culture, thus requiring another specimen to be collected and tested and adding cost and a delayed turn-around-time to the analysis. This is particularly important for some bacterial pathogens, in which culture for antimicrobial susceptibility testing and subtyping/strain identification is indicated. Readers are referred to several other articles for a further discussion of the advantages and disadvantages of NAATs for the diagnosis of acute diarrheal illness [38,51].

Unlike with *Cryptosporidium*, the genotyping of clinical isolates of *C. cayetanensis* is still relatively in its infancy. Genotyping is not needed for the clinical management of a patient but can be useful for trace-back investigations during outbreaks. Because *C. cayetanensis* undergoes sexual reproduction, the presence of heterozygous sequences and mixed genotypes may be expected [54]. The first method developed was multilocus sequence typing (MLST) based on microsatellite markers [54]. Originally, this was performed using products amplified from nested PCR [55]. However, results were often uninterpretable, possibly due to PCR products with different repeat lengths. Furthermore, nested PCR is laborious and may lead to the cross-contamination of amplicons [54]. Hofstetter et al. (2019) were able to improve the algorithm by eliminating the nested PCR step (although they suggest that nested PCR may be more sensitive in specimens with few oocysts) [54]. More recently, an ensemble-based distance statistic was evaluated using MLST with products derived from targeted amplicon deep sequencing (TADS). This TADS-MLST method facilitated genetic clustering with 93.8% sensitivity and 99.7% susceptibility and may greatly improve epidemiologic investigations during outbreaks [56]. Interestingly, whole-genome sequencing is not practical for the molecular surveillance of *C. cayetanensis* in outbreaks due to the large (at least 44 MB) genome, low DNA yield from stool specimens, and the lack of an animal model or culture system to enhance parasite yield [56,57].

## 12. Conclusions and Future Directions

It is clear that cyclosporiasis is no longer only a threat to individuals living in developing countries with poor sanitation. Annual summertime outbreaks have become a regular phenomenon in the United States, and cases have been linked to both imported and locally grown produce. Similarly, annual seasonal outbreaks linked to imports have been occurring for over 20 years in Canada. Meanwhile, ongoing outbreaks in highly endemic countries such as Mexico have resulted in increasing numbers of cases reported in travelers from the United Kingdom and Europe over the past several years [58]. The widespread adoption of rapid multiplex NAATs such as the BioFire GI pathogen panel has likely facilitated the recognition of infection and contributes to the growing numbers of reported cases each year. Similarly, molecular tools for *C. cayetanensis* genotyping have facilitated our understanding of the parasite’s transmission and distribution in the environment. Further research will allow for greater understanding of the genetic diversity of *C. cayetanensis*, its relationship to zoonotic *Cyclospora* species, and the risk it poses to our food and water supply.

## Figures and Tables

**Figure 1 microorganisms-09-01863-f001:**
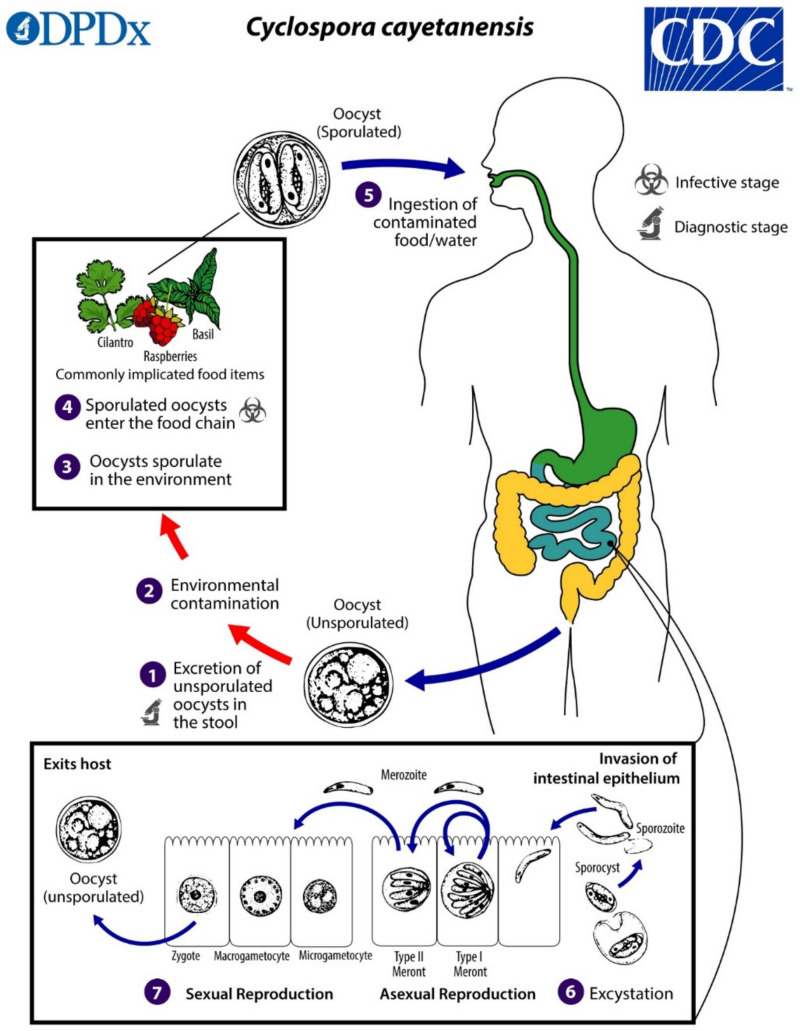
Life cycle of *Cyclospora cayetanensis*. Unsporulated, non-infective oocysts are passed in the feces (**1**). In the environment (**2**), sporulation occurs after days or weeks at temperatures between 22 °C to 32 °C, resulting in the division of the sporont into two sporocysts, each containing two elongated sporozoites (**3**). The sporulated oocysts can contaminate fresh produce and water (**4**) images, which are then ingested (**5**). The oocysts excyst in the gastrointestinal tract, freeing the sporozoites, which invade the epithelial cells of the small intestine (**6**). Inside the cells, they undergo asexual multiplication into Type I and Type II meronts. Merozoites from Type I meronts perpetuate the asexual cycle, while merozoites from Type II meronts undergo sexual development into macrogametocytes and microgametocytes upon invasion of another host cell. Fertilization occurs, and the zygote develops to an oocyst, which is released from the host cell and shed in the stool (**7**). Figure courtesy of the CDC-DPDx.

**Figure 2 microorganisms-09-01863-f002:**
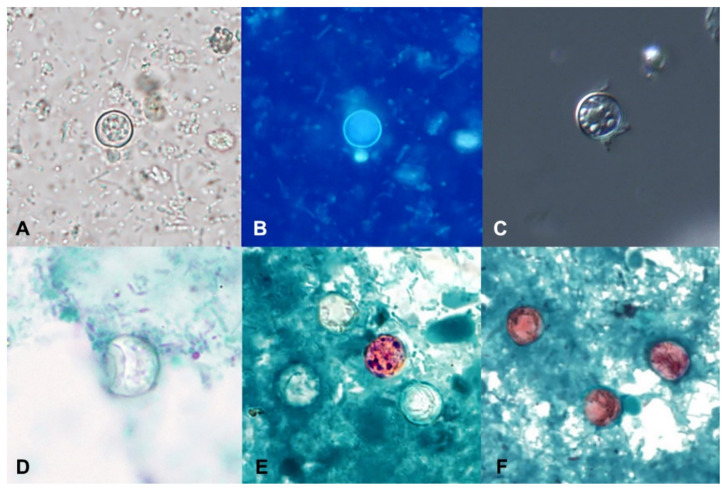
Oocysts of *Cyclospora cayetanensis* in stool specimens observed under different staining methods. (**A**) unstained concentrated wet mount. (**B**) UV autofluorescence. (**C**) differential interference contrast (DIC). (**D**) trichrome stain. (**E**) Kinyoun’s modified acid-fast. (**F**) modified safranin. (Figures courtesy of the CDC-DPDx).

**Figure 3 microorganisms-09-01863-f003:**
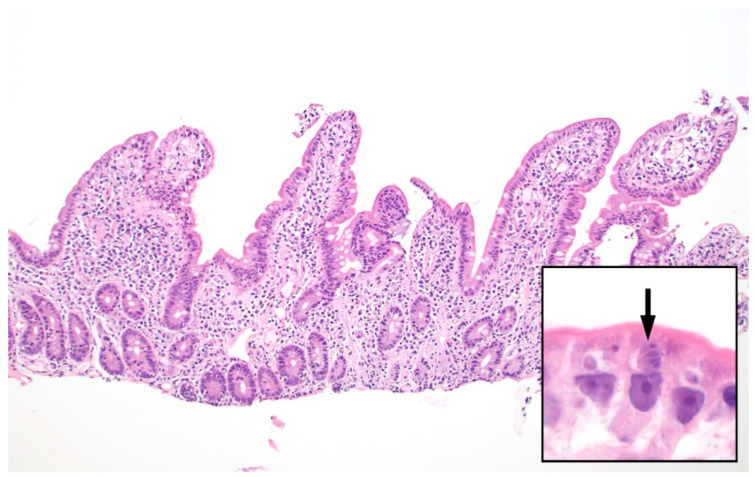
Small intestinal biopsy showing villous blunting and increased chronic inflammatory cells within the lamina propria (hematoxylin and eosin stain, 100x magnification). Higher power (inset, 1000×) shows intracellular *Cyclospora cayetanensis* parasites including a Type II meront containing 4 merozoites (arrow). Each merozoite shown here measures approximately 3–4 micrometers.

**Table 1 microorganisms-09-01863-t001:** Microscopic methods for the detection of *Cyclospora cayetanensis*.

Diagnostic Method	Advantages	Disadvantages
Stool Microscopy		
Direct wet mount	Fast, inexpensive; simultaneous detection of other intestinal parasites	Lack of sensitivity without concentration step; lack of defined morphologic features might make detection difficult for microscopists
Concentrated wet mount	Fast, inexpensive; simultaneous detection of other intestinal parasites	Lack of defined morphologic features might make detection difficult for microscopists
Differential Interference Contrast (DIC)	Increased sensitivity by highlighting internal structures	Not routinely available in many diagnostic labs
Ultraviolet autofluorescence	More sensitive than permanent smears; simultaneous detection of other coccidian oocysts and several helminth eggs	Requires specific UV filters that may not be routinely present in diagnostic labs
Lacto-phenol cotton blue	Fast, inexpensive; may be advantageous in resource-poor areas where acid-fast staining is not available	Non-specific; likely false positives with fungal elements
Trichrome/iron hematoxylin stain	Simultaneous detection of other intestinal protozoans	Oocysts do not stain with trichrome
Modified Ziehl-Neelsen (ZN) stain	Increased sensitivity over traditional O&P exams	Inconsistent staining of oocysts
Kinyoun’s modified acid-fast (MAF) stain	Increased sensitivity over traditional O&P exams	Inconsistent staining of oocysts
Modified safranin	More consistent staining of oocysts over ZN and MAF	Requires heating of stain
Auramine O (auramine-phenol)	More sensitive than traditional O&P exams	May be less sensitive than MAF, ZN; requires fluorescent microscope
**Histopathology**		
Hematoxylin-and-eosin (H&E), periodic acid Schiff (PAS)	Identify multiple developmental stages of *C. cayetanensis*	Not routinely ordered for *C. cayetanensis*; may be difficult to distinguish from *Cystoisospora belli*
Ziehl–Neelsen stain, Fite’s acid-fast stain	Can detect oocysts in tissues	Pre-oocyst stages may not stain

**Table 2 microorganisms-09-01863-t002:** Commercially available nucleic acid amplification tests (NAATs) for the detection of *Cyclospora cayetanensis*.

Assay	Manufacturer (Location)	Parasite and Microsporidial Targets	Approval *
BioFire^®^ FilmArray^®^ Gastrointestinal (GI) Panel	Biomérieux (Lyon, France)	*Cyclospora cayetanensis*, *Cryptosporidium* spp., *Giardia duodenalis*, *Entamoeba histolytica*	FDA, CE
Allplex™ Gastrointestinal Panel	Seegene (Seoul, South Korea)	*C. cayetanensis*, *Blastocystis* spp., *Cryptosporidium* spp., *Dientamoeba fragilis*, *E. histolytica*, and *G. duodenalis*	CE
QIAstat-Dx^®^ Gastrointestinal Panel	Qiagen (Hilden, Germany)	*Cyclospora cayetanensis*, *Cryptosporidium* spp., *Giardia duodenalis*, *Entamoeba histolytica*	CE
EasyScreen™ Enteric Protozoan Extended Detection Kit	Genetic Signatures (Newtown, Australia)	*C. cayetanensis*, *Blastocystis* spp., *Cryptosporidium* spp., *Dientamoeba fragilis*, *E. histolytica*, and *G. duodenalis*, *Enterocytozoon bieneusi*, *Encephalitozoon intestinalis*	CE
Novodiag^®^ Stool Parasites	Mobidiag (Espoo, Finland)	*Ancylostoma duodenale*, *Ascaris lumbricoides*, *Balantioides coli*, *Blastocystis* spp., *Clonorchis*/*Opisthorchis*/*Metorchis*, *Cryptosporidium* spp., *C. cayetanensis*, *Cystoisospora belli*, *D. fragilis*, *Dibothriocephalus* spp., *Encephalitozoon* spp., *E. histolytica*, *Enterobius vermicularis*, *E. bieneusi*, *Fasciola* spp., *Fasciolopsis buski*, *G. duodenalis*, *Hymenolepis diminuta*, *H. nana*, *Necator americanus*, *Schistosoma mansoni*, *Schistosoma* spp., *Strongyloides stercoralis*, *Taenia saginata*/*suihominis* (=*asiatica*), *T. solium*, *Trichuris* spp.	CE

* FDA, Food and Drug Administration; CE, Conformitè Europëenne.

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
