# Peer review of "Cyclosporiasis—Updates on Clinical Presentation, Pathology, Clinical Diagnosis, and Treatment"

_microorganisms, 2021, doi:10.3390/microorganisms9091863_

Round 1

Reviewer 1 Report

This paper is a well-written summary of cyclosporiasis.  I have but two suggestions for the authors to consider:

In section 6 - "Diagnosis" lines 157-159. I would argue that cyclosporiasis should be considered as a primary differential in any patient presenting with compatible symptoms with illness onset during the cyclosporiasis peak-period in the U.S. (typically May through August), not just those with a history of eating fresh leafy greens, berries, basil and cilantro.  In my experience, physicians don't often take the time to collect nutritional histories, and the timeframe of interest for cyclosporiasis (i.e. 2 weeks prior to illness onset) is much different than for other foodborne illnesses. 

Also consider mentioning that multiple stool specimens may be required to achieve diagnosis, as even persons with profuse diarrhea may be shedding few oocysts (which may be more applicable to microscopy).  

Author Response

Thank you for the helpful comments. We have edited our manuscript accordingly. 

Comment 1. We have changed the text to read "In the United States, where cyclosporiasis has become a seasonal illness in the summer, C. cayetanensis should also be considered as a primary differential in any patient presenting with compatible symptoms and illness onset during the cyclosporiasis peak period (i.e., May through August). A history of consuming fresh leafy greens, berries, basil, and cilantro within 2 weeks prior to illness onset may raise the clinical suspicion for cyclosporiasis, although this type of nutritional history is not commonly obtained 12"

Comment 2. We have added the following sentence to the section on diagnosis: Importantly, it may be necessary to examine multiple stool specimens for C. cayetanensis in order to make the diagnosis of cyclosporiasis, as that the number of oocysts shed in stool may be relatively few.

Reviewer 2 Report

Conventional clinical microbiology techniques aiming to establish etiologic agents both acute and chronic diarrheal illness due to gastrointestinal infection based on culture and immunoassays are limited by their low sensitivity and long turnaround time. While, NAATs for enteric pathogens allow for the syndromic testing of stool for multiple pathogens simultaneously and have higher sensitivity with a shorter turnaround time. Among enteric pathogens detected by NATTs is Cyclospora cayetanensis which begins to gain importance as more and more epidemics associated with it concern non-endemic regions.

In peer-reviewed manuscript ‘Cyclosporiasis – Updates on Clinical Presentation, Pathology, Clinical Diagnosis, and Treatment’ Authors in detail described advantages and limitations of both commonly used diagnostic methods, most often microscopic, and gradually introduced molecular methods, including NATTs in  C. cayetanensis detection and recognition. The review article is a solid portion of knowledge that can be the basis for new research in this direction. What should be emphasized is a critical look at the issue presented. In addition, the theme is conveyed in a clear way.

Nevertheless, I would appreciate a small addition. I would recommend that you discuss the limitations of using NATTs in more detail, not just technical and economical aspects. After signaling such problems, an alternative is to refer the reader to, for example, another review article, where it is well described for example:

(Yalamanchili H, Dandachi D, Okhuysen PC. Use and Interpretation of Enteropathogen Multiplex Nucleic Acid Amplification Tests in Patients With Suspected Infectious Diarrhea. Gastroenterol Hepatol (N Y). 2018 Nov;14(11):646-652) or another one chosen by the Authors.

Author Response

Thank you for the suggestion to elaborate on the pros and cons of NAATs for detection of cyclosporiasis. We have expanded our paragraph in the section on molecular diagnostics to further elaborate on the pros and cons of NAATs as follows: "The use of molecular tools could speed up recognition of an outbreak. Because outbreaks are usually associated with ephemeral food sources (raw, fresh fruits and vegetables), by the time an outbreak is realized, the source may have been consumed or discarded, making trace-back investigations difficult50. NAATs also do not require parasitology expertise, and some are available in an easy-to-use, cartridge-based format that provide a result in as little as 1 hour. Yet, despite the increased sensitivity and specificity afforded by NAATs, there are some disadvantages associated with molecular diagnosis. In general, NAATS are still the most expensive option for routine clinical diagnosis and require sophisticated instruments and proprietary reagents. Also, some of the commercially available tests, as well as most LDTs, for C. cayetanensis must be performed in a high complexity laboratory with strict containment controls and unidirectional workflows. Finally, it is important to note that NAATs can detect DNA from both viable and dead organisms, and thus should not be used as a test of cure, as DNA may continue to be shed in the stool for some time after successful treatment 51. Similarly, detection of C. cayetanensis DNA does not necessarily indicate that the organism is the cause of the patient’s illness, as infection with this parasite may be asymptomatic and there may be another responsible pathogen. Finally, specimens submitted for NAAT may not be compatible with additional tests such as microscopy and culture, thus requiring another specimen to be collected and tested and adding cost and delayed turn-around-time to the analysis. This is particularly important for some bacterial pathogens, in which culture for antimicrobial susceptibility testing and subtyping/strain identification is indicated. Readers are referred to several other articles for a further discussion of the advantages and disadvantages of NAATs for diagnosis of acute diarrheal illness." (suggested article and 2 others added here)

Reviewer 3 Report

This is an interesting review and enjoyable read. I have only a few minor comments and edits.

Line128; Add one, to read, ≥ one month …

Line 157; Annual seasonal outbreaks have been occurring for over 20 years in Canada, linked to imported produce. Thus, I suggest you add ‘and Canada’ to read In the United States and Canada, where ….

Line 173; delete a

Line 282; change in to an to read, an increase in the number ....

Line 295; either change does to do, or make publication singular

Line 310, should be NAATs

Line 314; remove I, to read, Finally, it is....,

Line 339; Annual seasonal outbreaks have been occurring for over 20 years in Canada, linked to imported produce.

Lines 308-317; When discussing some of the disadvantages, I think it is important to note here that the preservative used for stool specimens impacts molecular diagnostics. Formalin-based fixatives, such as Sodium Acetic Acid Formalin (SAF), are not suitable for PCR due to degradation of DNA, however SAF is still being used in some countries, including frequently in Canada.

Author Response

Thank you for the thorough review of our manuscript, and for catching several grammatical errors that we made throughout. We have corrected all of these as suggested by the author. We also added a reference to Canada as requested, and in our final conclusions: " Annual summertime outbreaks have become a regular phenomenon in the United States, and cases have been linked to both imported and locally grown produce. Similarly, annual seasonal outbreaks linked to imported have been occurring for over 20 years in Canada."